biomonitoR: an R package for managing ecological data and calculating biomonitoring indices

Laini Alex 1 2
Guareschi Simone s.guareschi@ebd.csic.es 3 4
Bolpagni Rossano 1
Burgazzi Gemma 5
Bruno Daniel 6
Gutiérrez-Cánovas Cayetano 3
Miranda Rafael 7
Mondy Cédric 8
Várbíró Gábor 9
Cancellario Tommaso 7 10
1 Department of Chemistry, Life Sciences and Environmental Sustainability, University of Parma , Parma , Italy
2 Department of Life Sciences and Systems Biology, University of Turin , Torino , Italy
3 Estación Biológica de Doñana, Consejo Superior de Investigaciones Científicas , Sevilla , Spain
4 Geography and Environment, Loughborough University , Loughborough , United Kingdom
5 Institute for Environmental Sciences, University of Koblenz-Landau , Landau , Germany
6 Instituto Pirenaico de Ecología (IPE), Consejo Superior de Investigaciones Científicas , Zaragoza , Spain
7 Department of Environmental Biology, University of Navarra , Pamplona , Spain
8 French Agency for Biodiversity (OFB) , Vincennes , France
9 Centre for Ecological Research, Institute of Aquatic Ecology , Debrecen , Hungary
10 Water Research Institute, National Research Council (CNR) , Verbania , Italy
Ward Eric
Electronic publication date: 2022 Oct 14
Publication date: 2022
Volume: 10
Electronic Location ID: e14183
Received 2022 Jul 22; Accepted 2022 Sep 14
Copyright: ©2022 Laini et al.
Copyright year: 2022
Copyright holder: Laini et al.
License: This is an open access article distributed under the terms of the Creative Commons Attribution License, which permits unrestricted use, distribution, reproduction and adaptation in any medium and for any purpose provided that it is properly attributed. For attribution, the original author(s), title, publication source (PeerJ) and either DOI or URL of the article must be cited.
License URL: https://creativecommons.org/licenses/by/4.0/

Keywords: Bioassessment, Decision-making tools, Ecological indicators, Environmental management, Functional metrics, Taxonomic indices

Funding: COST (European Cooperation in Science and Technology) action CA15113 (Science and Management of Intermittent Rivers and Ephemeral Streams) PRIN-NOACQUA (Community responses and ecosystem processes in intermittent streams, Prot. 201572HW8F) CSIC Interdisciplinary Thematic Platform (PTI) Síntesis de Datos de Ecosistemas y Biodiversidad (PTI-ECOBIODIV) “Juan de la Cierva—Incorporación” MINECO, IJC2018-036642-I Royal Society-Newton International Fellowship at Loughborough University (NIF R1 180346) European Regional Development Fund (SUMHAL, LIFEWATCH-2019-09-CSIC-13, POPE 2014–2020) at Doñana Biological Station (2021–2022) National Research, Development and Innovation Office NKFIH FK 135 136 Alex Laini was supported by COST (European Cooperation in Science and Technology) within the Action CA15113 (Science and Management of Intermittent Rivers and Ephemeral Streams) and by the project PRIN-NOACQUA (Community responses and ecosystem processes in intermittent streams, Prot. 201572HW8F). Daniel Bruno was funded by CSIC Interdisciplinary Thematic Platform (PTI) Síntesis de Datos de Ecosistemas y Biodiversidad (PTI-ECOBIODIV). Cayetano Gutiérrez Cánovas was supported by a “Juan de la Cierva—Incorporación” contract (MINECO, IJC2018-036642-I). Simone Guareschi was supported by a Royal Society-Newton International Fellowship at Loughborough University (NIF R1 180346) and by European Regional Development Fund (SUMHAL, LIFEWATCH-2019-09-CSIC-13, POPE 2014–2020) at Doñana Biological Station (2021–2022). Gábor Várbíró was supported by the National Research, Development and Innovation Office—NKFIH FK 135 136 grant. There was no additional external funding received for this study. The funders had no role in study design, data collection and analysis, decision to publish, or preparation of the manuscript.

==============================
The monitoring of biological indicators is required to assess the impacts of environmental policies, compare ecosystems and guide management and conservation actions. However, the growing availability of ecological data has not been accompanied by concomitant processing tools able to facilitate data handling and analysis. Multiple common challenges limit the usefulness of biomonitoring information across ecosystems and biological groups. Biomonitoring data analysis is currently constrained by time-consuming steps for data preparation and a data processing environment with limited integration in terms of software, biological groups, and protocols. We introduce biomonitoR, a package for the R programming language that addresses technical challenges for the management of ecological data and metrics calculation. biomonitoR implements most of the biological indices currently used or proposed in different fields of ecology and water resource management. Its combination of customizable functions aims to support a transferable and comprehensive biomonitoring workflow in a user-friendly environment. biomonitoR represents a versatile toolbox with five main assets: (i) it checks taxonomic information against reference datasets allowing for customization of trait and sensitivity scores; (ii) it supports heterogeneous taxonomic resolution allowing computations at multiple taxonomic levels; (iii) it calculates multiple biological indices, including metrics for both broad and stressor-specific ecological assessments; (iv) it enables user-friendly data visualization, helping both decision-making processes and data interpretation; and (v) it allows working with an interactive web application straight from R. Overall, biomonitoR can benefit the wide biomonitoring community, including environmental private consultants, ecologists and natural resource managers.

Introduction

The Anthropocene has brought multiple and emerging threats to natural ecosystems and their biological communities (Reid et al., 2019). The monitoring of biological indicators (so called “biomonitoring”) is crucial to evaluate the impacts of environmental policies, compare ecosystems and guide management and conservation actions (Goldsmith, 1991; Jackson et al., 2016). In this context, biomonitoring represents a leading example of how ecological knowledge can help to address broad societal issues (Friberg, 2014). Over the years, research and institutional activities (e.g., long-term monitoring observatories, citizen science programmes) as well as the growing capacity for data storage (e.g., international platforms like Global Biodiversity Information Facility), have facilitated access to an increasing amount of ecological and biological data covering large areas, time frames and taxonomic groups (e.g., Grenié et al., 2022). Such heterogeneous sources and data availability brought new challenges and have not been adequately coupled with corresponding processing tools able to fully explore the great potential of biological databases, especially from a monitoring perspective.

Due to their long-standing exposure to human exploitation (Feio et al., 2021; Friberg, 2014), the case of freshwater ecosystems is emblematic. A multitude of information has been produced and indices have been developed for freshwater systems, especially following the implementation of environmental legislation such as the Water Framework Directive in Europe, the Clean Water Act in USA or the Canadian Protection Act (Birk et al., 2012; Vitecek, Johnson & Poikane, 2021). In this case, biomonitoring relies on a set of biological elements such as phytoplankton, phytobenthos, macrophytes, macroinvertebrates, and fish for assessing the status and integrity of communities and ecosystems (Birk et al., 2012). Complexity arises due to the different type of metrics used in freshwater biomonitoring including diversity measures (e.g., taxa richness, Shannon index) and biological indices that rely on taxon-specific sensitivity (Buss et al., 2015). Functional trait-based indices (e.g., Mondy et al., 2012; Laini et al., 2022) also emerged as novel tools able to provide complementary information and potentially enhanced performance over traditional indices (Bruno et al., 2016; Cadotte, Carscadden & Mirotchnick, 2011, but see Wilkes et al., 2017 for an exception). Moreover, new and consolidated threats (e.g., biological invasions, climate change) call for further pressure-based indices, like biocontamination (Arbačiauskas et al., 2008) and flow intermittency (Chadd et al., 2017) metrics.

In this context, an integrative data processing environment for biomonitoring is still lacking and would facilitate data-processing and index calculation, which are currently poorly integrated in terms of specific national protocols and informatic programs. One of the most important data-processing challenges is addressing the inconsistency among taxonomic, trait, and sensitivity scores because mismatches can lead to excluding taxa from analyses (Guareschi & Wood, 2019). Likewise, for highly diverse taxa (e.g., invertebrates), data frequently relies on mixed taxonomic resolution (e.g., species, genus, and family) due to inherent difficulties in identifying taxa at species level (e.g., in early life stages) (Jones, 2008). Heterogeneous taxonomic resolution and nomenclature discrepancies complicate assigning sensitivity scores or trait values to taxa and requires time-consuming data manipulations to properly organize the dataset. A thorough exploitation of these data, which are often only partially explored, will optimize the use of ecological data and reduce inefficiencies (Patterson et al., 2010; Grenié et al., 2022). Similarly, functional indices calculated from trait-based information require complex calculations that may prevent their wide use in biomonitoring (Maire et al., 2015).

Here, we introduce biomonitoR, a new R package (R Core Team, 2021) supporting the calculation of a wide range of biomonitoring indices and the effortless management of biological information from different sources (Fig. 1). The package favours a smooth and reproducible workflow in biomonitoring science and allows both new operations (e.g., metrics calculation and traits manipulation) as well as existing ones currently requiring the use of several packages. This versatility will help biomonitoR gain a prominent position among the informatic tools in ecology and monitoring. A detailed comparison with popular R packages focusing on community data analysis is illustrated in Table 1, displaying the niche occupied by biomonitoR within them.

Materials & Methods

Import data

The first step consists in comparing user’s data to a reference taxonomic dataset with the function as_biomonitor. User’s data consists in a data.frame with taxa on the first column and occurrences/abundances on the following columns, while the taxonomic reference dataset includes the relationships of taxa among levels. biomonitoR comes with four built-in reference datasets for diatoms, macrophytes, macroinvertebrates and fish derived from Diat.barcode (Rimet et al., 2019), AlgaeBase (Guiry et al., 2014), freshwatercology.info (Schmidt-Kloiber & Hering, 2015), and FishBase (Froese & Pauly, 2019), respectively. To ensure an up-to date and global taxonomy information, several functions (e.g., get_gbif_taxa_tree) are already available, permitting the user to build a biomonitoR reference dataset from online resources such as the Global Biodiversity Information Facility (GBIF), World Register of Marine Species (WoRMS), National Biodiversity Network Atlas (NBN Atlas) and International Union for Conservation of Nature (IUCN) (SM1). Moreover, users can also provide their own reference dataset or build it from a taxonomic tree with the function ref_from_tree. This is a crucial step in data handling to organize and clean the dataset as well as unify writing styles (e.g., lexical variants). Taxa names that do not match with any taxon of the reference taxonomy are excluded to ensure consistency of further steps. Nonetheless, suggestions are proposed either silently or interactively by setting the correct_names argument to TRUE. To explore the general data structure, the object generated by as_biomonitor can be subsetted or plotted by leveraging on the interactive plotly package (Sievert, 2020). The second step consists in using the function aggregate_taxa to aggregate taxa at different taxonomic levels according to the user’s data resolution. The obtained object can be used to calculate all the indices implemented in biomonitoR at the desired taxonomic resolution.

Figure 1 Example of workflow in environmental and biomonitoring science (from field work, data manipulation to results dissemination).

biomonitoR represents the toolbox that supports biomonitoring tasks and fluently connects the different phases.

Table 1 Comparison between the biomonitoR package and other R packages.

Comparison between the biomonitoR package and four popular R packages for analysing ecological community data. Package versions and dates are also specified.

Tasks and measures	vegan 2.6-2 2022-04-17	FD 1.0-12.1 2022-05-02	taxize 0.9.100 2022-04-22	BiodiversityR 2.14-3 2022-08-06	biomonitoR 0.9.3 2022-06-13	
Richness and diversity metrics	Yese.g., diversity, fisher.alpha, specnumber, simpson.unb, taxondive, tsallis	No	No	Yese.g., diversityresults	Yes allrich, berpar, brillouin, esimpson, fisher, get_taxa_abundance, get_taxa_richness, invberpar, invsimpson, mcintosh, margalef, menhinick, pielou, richness, shannon, simpson	
Biomonitoring indices	No	No	No	No	Yes aspt, bioco, bmwp, dehli, epsi, ept, eptd, fuzzy_trait_ratio, get_taxa_abundance, get_taxa_richness, ibmr, igold, life, psi, whpt,	
Functional indices	Yese.g., treedive	Yese.g., dbFD, fdisp	No	No	Yes csi, cwm, f_disp, f_divs, f_red, f_eve, f_rich, fuzzy_trait_ratio	
Comparison with reference dataset, handling heterogeneous taxonomic resolution	No	No	Yese.g., classification	No	Yes as_biomonitor, get_gbif_taxa_tree, get_iucn_taxa_tree, get_nbn_taxa_tree, get_worms_taxa_tree	
Managing trait information	No	No	No	No	Yes assign_traits, average_traits, manage_traits, sample_traits	
Ordinations, clustering	Yese.g., cca, rda, metaMDS	No	No	Yese.g., CAPdiscrim	No	
Beta diversity analysis	Yese.g., betadiver	No	No	No	No	
Null modelling	Yese.g., hiersimu, commsim	No	No	No	No	

Multiple taxonomic levels and back trace

biomonitoR calculates ecological indices at different taxonomic levels. The desired taxonomic level (e.g., species, family) can be specified with the argument tax_lev. This is a key feature to develop robust methods based on taxonomic sufficiency, the level of taxonomic detail to which organisms must be identified to recognize ecological patterns (Jones, 2008).

To keep track of the computations performed by biomonitoR, the original information can be obtained by setting the traceB argument to TRUE. This is especially useful when mismatches between taxonomic and sensitivity/traits datasets lead to the exclusion of some taxa. To further assure the transparency of the entire process, default scores are made available by using the show_scores function. biomonitoR is designed to be flexible, such that users can use their own scores or trait information. For example, in river biomonitoring, the calculation of the internationally used Average Score Per Taxon index (ASPT; Hawkes, 1998, see SM2) can be adapted to user-defined scores and taxonomic aggregation rules (e.g., combining taxa with similar stressor-specific responses). This feature can be extremely helpful when users want to test indices sensitivity or for simulation studies.

Tools for managing trait-based information and indices availability

biomonitoR allows users to overcome problems frequently arising during the computation of trait-based indices. The package includes several functions to manage functional traits information allowing the calculation of a range of functional metrics most of which are based on the FD (Laliberté & Legendre, 2010) and ade4 packages (Dray & Dufour, 2007). For example, the function assign_traits matches the taxa to those of the traits dataset. This is particularly useful when traits of several taxa in the trait dataset (e.g., species- and genus-level) can be assigned to a higher taxonomic level (e.g., family level). Currently, biomonitoR comes with a built-in macroinvertebrate trait dataset (European fauna: Schmidt-Kloiber & Hering, 2015; Tachet et al., 2010) but users can also add their own (see an example of loading a dataset including North American insect traits in SM3). A finer control over trait information can be achieved with the manage_traits function, that allows users to select the traits belonging to the nearest taxon in the taxonomic tree (e.g., SM3). Finally, traits at fine levels (e.g., species or genus) can be averaged at coarser levels (family) (average_traits) or a random taxon (e.g., genus) can be selected as representative (sample_traits). The calculation of some functional indices (e.g., functional richness, dispersion and evenness) requires summarizing trait variation into a low dimensional trait space. This can be achieved through a principal coordinate analysis based on a matrix of Euclidean distances or Gower dissimilarities. The quality of the trait space can be evaluated and adjusted using the function select_pcoa_axes to avoid poor representations (Maire et al., 2015). This function implements three ways of evaluating the quality of the functional space: (i) the correlation between the Euclidean distance of the n selected axis and the overall distance; (ii) the r2 proposed by Legendre & Legendre (2012) and (iii) the mean squared deviation approach proposed by Maire et al. (2015). If the species-by-species distance matrix is not Euclidean, select_pcoa_axes calculates the performance of four widely used transformations (Cailliez, Lingoes, square root and quasi-Euclidean; see Legendre & Legendre, 2012). Sometimes, data transformation is not enough to make the species-by-species matrix Euclidean (e.g., when two or more taxa share the same traits) with potential negative effects on downstream data analysis. To resolve this issue, biomonitoR comes with two approaches. The first option aggregates the abundance of taxa with the same traits (zerodist_rm). The second option (add_bias_to_traits) differentiates taxa with identical trait profiles by adding a small amount (random bias) to the traits of each taxon (see R code and examples in SM3).

Overall, both taxonomic-based indices and more novel measures still pending to be fully incorporated in national or international biomonitoring schemes are available in the package. biomonitoR currently allows the calculation of more than 30 indices (including diversity, biomonitoring, and functional measures) as well as a wide range of complementary taxonomic measures (e.g., richness or abundance of a specific taxon, pair and combination of taxa, see Table 2 and examples of applications in SM2).

Data visualization

biomonitoR relies on the plotly package for providing flexible and high-quality graphs. Three types of plots are currently available to visualize the data. The first is a Sankey diagram that links indicator taxa (De Cáceres & Legendre, 2009) from a desired to an upper taxonomic level through groups provided by the user or identified by a cluster analysis (e.g., Fig. 2). The second plots the prevalence of a taxonomic level (e.g., family) within an upper taxonomic level (e.g., order, see SM4a-c). The third is an interactive barplot reporting the proportion of taxa within a desired taxonomic level, with sites ordered according to the results of a cluster analysis (see examples and codes in SM4d).

Table 2 Summary of the main indices implemented in the biomonitoR package. Full details about references, acronyms and R functions are available in Table SM2.

Index type	Index name	Target group	
Taxonomic-based indices (richness, diversity and evenness)	Richness or abundance of a taxon or combination of taxa, Berger-Parker, Berger-Parker inverse, Brillouin, Fisher alpha, Margalef, McIntosh, Menhinick, Pielou, Shannon, Simpson, Simpson evenness, Simpson inverse	All biological groups	
Biomonitoring indices (ecological and single pressure assessments)	ASPTa, BMWPa, WHPTa, PSIa, EPSIa, LIFEa, Flow-T, DEHLI, EPT, log10(SEL_EPTD +1), 1-GOLD, RCI, ACI, SBCI, IBMR	Macroinvertebrates, Macrophytes, other biological groups	
Functional trait-based indices	Functional Richness, Diversity, Redundancy, Evenness and Dispersion, Community trait specialization index, Taxon Specialization Index, Community Weighted Mean	All biological groups	
Notes.

a These indices can be calculated using the composite family approach that aggregates families having similar ecological requirements. Moreover, biomonitoR provides, by default, four sets of sensitivity scores for ASPT and BMWP (two developed UK, one in Italy and one in Spain) and two sets of sensitivity scores for LIFE (developed in UK).

Figure 2 Example of Sankey plot application (function plot_indicator_taxa) showing indicator taxa for an aquatic insect community in intermittent and perennial river sites.

Orders are listed on the left and families/genera on the right.

Results & Discussion

Package installation and basic usage example

The biomonitoR package (version 0.9.3) is available on GitHub (https://github.com/alexology/biomonitoR). The package can be installed as follows:

library(devtools)

install_github(“alexology/biomonitoR”, ref = “main”, build_vignettes = TRUE)

An R Shiny implementation (Chang et al., 2021), as interactive app of the package, is available from the R console running the command line:

library(shiny)

runGitHub(“biomonitoR_app”, “TommasoCanc”, ref = “main”, subdir = “biomonitoR_app”)

as well as in the webpage (https://tcanceco.shinyapps.io/biomonitoR_app/).

These complementary options allow working with a user-friendly graphical interface that will benefit different audiences (either familiar or not with R programming language), thus making biomonitoR widely accessible and interactive.

Usage example

# Load the package

library(biomonitoR)

# Load one of the built-in macroinvertebrate datasets (mi_prin)

data(mi_prin)

# Details about the dataset

?mi_prin

# Import data in the biomonitoR format

data_asb <- as_biomonitor(mi_prin, group = “mi”)

# Inspect the overall community structure by plotting the object

plot(data_asb)

# Aggregate taxa at different taxonomic levels

data_agg <- aggregate_taxa(data_asb)

# Calculate a taxonomic index

# e.g., Shannon index at taxa and family level

shannon(data_agg, tax_lev = “Taxa”)

shannon(data_agg, tax_lev = “Family”)

# Calculate a biomonitoring index

# e.g., ASPT index with function aspt (see ?aspt for details)

aspt(data_agg)

Detailed scripts for running further analyses (e.g., data manipulation, biomonitoring indices calculations, function customization, plots) are provided in SM2-3-4. The package is being regularly updated and we encourage interested users to look for the latest version in GitHub and to report any suggestion for further development.

Limitations and perspectives

Despite its flexibility, biomonitoR may presents some punctual limitations. The reference dataset needs to be implemented according to the taxonomic levels that are currently available in the package. This can be an issue in the occasional cases where species-groups, life stage information or operational taxonomic units are needed. With minor adjustments (e.g., considering species groups as species) their use is still possible with the current implementation. Moreover, according to Tables 1–2, one of the main biomonitoR assets (i.e., multiple index calculation) is biased towards the biomonitoring of running waters. This is due to the unparalleled pressures affecting freshwater biodiversity worldwide and the long history of biological monitoring in rivers, that date back to the beginning of the 20th century (Bonada et al., 2006). Nonetheless, a detailed biomonitoring toolbox, not constrained at local scale, is still missing (see Table 1) and biomonitoR aspires to fill this gap. It should be noted that, despite the primary focus at European level, the package can potentially achieve a more international perspective after further customizations. Most of the river biomonitoring metrics tested in Africa, South-Central America, Mexico, Australia, and Asia are, in fact, directly related to the Biological Monitoring Working Party scoring systems (BMWP/ASPT, Aschalew & Moog, 2015; Eriksen et al., 2021; Feio et al., 2021) originally developed in the UK and already available in biomonitoR (Table 2).

Regardless of the primary freshwater focus, the package can instantly manage biological data from other domains (e.g., marine, terrestrial) and communities (e.g., non-aquatic plants, vertebrates) if a reference dataset is uploaded by the users or specifically built from online sources with the internal functions. In addition, general-purpose indices for biodiversity analysis and assessment (e.g., taxonomic and functional measures) can be already computed.

Conclusions

Rigorous and updated biomonitoring procedures and tools are crucial for biodiversity conservation research (e.g., early warning, long-term assessment, and observatories) and to properly inform science-based policies. Here we presented biomonitoR, a new R tool that creates the basis for a common environment for dataset management that allows users to be in an advantageous position in research and applied fields related to environmental science.

The set of functions, available in a single open access package, support task automation and greatly facilitate repeatability and reproducibility among studies. Therefore, biomonitoR may also ensure greater consistency among biomonitoring programs which could lead to more far-reaching analyses at spatial and temporal scales.

The package represents a flexible toolbox for (i) checking taxonomic, sensitivity and traits information against a reference taxonomic database; (ii) managing heterogeneous taxonomic resolution and computing indices at multiple taxonomic levels; (iii) calculating metrics and biological indices for both general and stressor-specific ecological assessment; (iv) interpreting results through interactive data visualization; (v) working with a user-friendly web application directly available in R.

Overall, biomonitoR enables high-performing options for better management and harmonization of ecological and biomonitoring data while facilitating the interpretation and dissemination of results. Therefore, the package has significant value in decision-making processes and benefits the wide biomonitoring community.

Supplemental Information

Supplemental Information 1 R file with usage examples

Example code for community data analyses with the biomonitoR package

Supplemental Information 2 Reference datasets in biomonitoR and list of the main metrics available in biomonitoR

Supplemental Information 3 Default plots in biomonitoR: Examples and code

This package is based upon work from COST Action CA15113 (SMIRES, Science and Management of Intermittent Rivers and Ephemeral Streams, http://www.smires.eu). Authors would like to thank the Alpine Stream Research Centre (ALPSTREAM, Ostana, Italy), Roberto Mattioli for his help in preparing Fig. 1 and Anna-Maria Sourelli for English grammar check.

Additional Information and Declarations

Competing Interests

Author Contributions

Data Availability

The authors declare there are no competing interests.

Alex Laini conceived and designed the experiments, performed the experiments, analyzed the data, prepared figures and/or tables, authored or reviewed drafts of the article, and approved the final draft.

Simone Guareschi performed the experiments, analyzed the data, prepared figures and/or tables, authored or reviewed drafts of the article, and approved the final draft.

Rossano Bolpagni performed the experiments, authored or reviewed drafts of the article, and approved the final draft.

Gemma Burgazzi performed the experiments, authored or reviewed drafts of the article, and approved the final draft.

Daniel Bruno performed the experiments, analyzed the data, authored or reviewed drafts of the article, and approved the final draft.

Cayetano Gutiérrez-Cánovas performed the experiments, analyzed the data, authored or reviewed drafts of the article, and approved the final draft.

Rafael Miranda performed the experiments, authored or reviewed drafts of the article, and approved the final draft.

Cédric Mondy performed the experiments, analyzed the data, authored or reviewed drafts of the article, and approved the final draft.

Gábor Várbíró performed the experiments, analyzed the data, authored or reviewed drafts of the article, and approved the final draft.

Tommaso Cancellario conceived and designed the experiments, performed the experiments, analyzed the data, authored or reviewed drafts of the article, and approved the final draft.

The following information was supplied regarding data availability:

The code and data are available on GitHub at https://github.com/alexology/biomonitoR.

It is also available at Zenodo: Laini, A., Guareschi, S., Bolpagni, R., Burgazzi, G., Bruno, D., Gutiérrez- Cánovas, C., Miranda, R., Mondy, C., Várbíró, G., & Cancellario, T. (2022). biomonitoR: an R package for managing ecological data and calculating biomonitoring indices. (v0.9.3). Zenodo https://doi.org/10.5281/zenodo.7074237.

The examples are available in the Supplemental Files.

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
