# Peer review of "biomonitoR: an R package for managing ecological data and calculating biomonitoring indices"

_PeerJ, doi:10.7717/peerj.14183_

## Round 0.1 · original submission · Minor Revisions

Thanks for your work on the manuscript, and your work putting a useful R package together. The reviewers have included a few minor comments / editorial changes that will improve the manuscript (see especially Reviewer 1 / 2 comments)

Reviewer 1 ·

Basic reporting

1.1 Very clear and concise abstract and main text
1.2 Line 52 should read “effectiveness of the IMPACT of environmental policies” (also in the abstract)
1.3 Line 72: A lot of published evidence suggests that trait-based indices do not perform well compared to taxonomy-based indices (e.g. Mckenzie et al., 2022) or that traits are weakly related to taxon sensitivity (e.g. Wilkes et al., 2017). I do not think you need to go into the possible reasons for this (i.e. limitations of available trait data), but the text should acknowledge that trait-based biomonitoring has not made much of an impact. Perhaps it would be sufficient to say that trait-based indices “provide complementary information and potentially enhanced performance over traditional indices”.
1.4 Line 224: Should be “not CONSTRAINED at A local scale”.

McKenzie, M., England, J., Foster, I. and Wilkes, M., 2022. Evaluating the performance of taxonomic and trait-based biomonitoring approaches for fine sediment in the UK. Ecological Indicators, 134, p.108502.

Wilkes, M.A., Mckenzie, M., Murphy, J.F. and Chadd, R.P., 2017. Assessing the mechanistic basis for fine sediment biomonitoring: Inconsistencies among the literature, traits and indices. River Research and Applications, 33(10), pp.1618-1629.

Experimental design

2.1 You have made a comprehensive package with some wonderfully useful functions. The manage_traits, average-traits and sample_traits are particularly practical.
2.2 Line 60: While it is fair to say that data processing tools lag behind the availability and heterogeneity of the data, it is less fair to claim that they “have not been coupled”. There are a very large number of software packages that deal specifically with processing of ecological data for different purposes. These include rgbif for downloading biological records, taxize for retrieving taxonomic information, spOccupancy for species occupancy modelling, sdm and biomod2 for species distribution modelling, and FD for functional diversity analyses, to name just a few. Your Table SM1 mentions further packages. In this part of the introduction, it would be better to acknowledge the breadth of packages available and show how biomonitoR fills an important gap.
2.3 Line 227: For reference, could you give some brief examples of how “a reference dataset is uploaded by the users or specifically built from online sources with the internal functions”. What are the functions involved in doing this? Adding some extra detail here may assist in achieving impact for the package beyond the freshwater realm.

Validity of the findings

3.1 Conclusions are well stated and a good level of detail has been provided in the main text and supplementary material.

Reviewer 2 ·

Basic reporting

The majority of the manuscript is presented in professional English, but there are a few instances that require editing or correction to improve grammar or flow. I have outlined some recommendations below:

Line 26: “required” instead of “crucial”
Line 27: “ecological data has not been accompanied by concomitant processing tools….”
Line 28: Do you need “may” in this statement?
Line 30: “fragmented in specific software, biological groups and national protocols”. This phrase isn’t clear to me. I don’t believe the biological groups or perhaps the national protocols are fragmented. There is certainly a lack of integration, but I don’t think fragmented is the right word choice.
Line 36: “dissemination of results”
Line 51: “required” instead of “crucial”
Line 53: “broad” instead of “wide”
Line 55: “have facilitated access to an increasing….time frames and taxonomic groups”
Line 59: “have not been adequately coupled with corresponding…”
Line 62: “Exploitation and scarcity”; I don’t disagree with this statement, but it might be appropriate to include references.
Line 63: “developed for freshwater systems, especially…”
Line 66: “In this case, freshwater biomonitoring …”
Line 76: “lacking” instead of “missing”; because elsewhere you indicate that some of these calculations can be done with other R packages
Line 85: “a better system would optimize the use…
Line 106: Can you clarify what you mean by “writing styles”? Do you mean formatting?
Line 132: “biomonitoR allows users….”
Line 133: “most of which are based on FD…”
Line 140: “that allows users to select…”
Line 155: resolve instead of solve
Line 176: follows
Line 181: “an interactive app…., is available from the R console…”
Line 223: Please check verb tenses in this sentence.
Line 224: “if” instead of “in case”
Line 232: “Multiple biodiversity conservation purposes” seems vague and perhaps awkward, simply provide the list you include parenthetically.
Line 234: “for dataset management that allows users”; can you clarify the last phrase in this statement? Can you be more specific about what researcher will be able to do and how this is an advantage? I am not doubting the potential, but the explanation falls short.
Line 236: “It” is a false subject. I suggest “The package” as a more specific alternative. Additionally, “disseminating results” would be more consistent with the rest of the points in this sentence (checking, managing, calculating, disseminating)
Line: 242. “…options for better management and ….. biomonitoring data while facilitating the dissemination and presentation of results.”

This manuscript doesn’t explicitly follow the traditional structure of a scientific paper, but the presentation is reasonable given the scope of work. The references are appropriate, but may not include relevant material for North America (see validity of the findings). Additionally, I believe at least one table (Table SM1) presented as supplemental may merit inclusion in the manuscript itself (see validity of the findings). Furthermore, since the primary product or outcome of this work is a collection of R files, I would recommend the authors consider compiling a complete R package for submission to the CRAN repository (see validity of the findings). I have elaborated on these recommendations in the other sections of this review.

Experimental design

Even though this manuscript lacks the typical structure of a scientific journal article, some aspects of the material could be interpreted as analogous to experimental design. Specifically, I would encourage the authors to elaborate on or further explain what is meant by “random bias” in Line 155 and the assumptions associated with applying trait data based on the nearest taxon in the taxonomic tree described in Line 140. Users would benefit from a clarification of these points in the narrative. Additionally, the authors are encouraged to consider including a brief description of the help files and example datasets (e.g., number of taxa and or records) available in the text of the manuscript. This would be useful not only to practitioners, but also to instructors that might use this package in upper level undergraduate and graduate courses.

Validity of the findings

Similar to the experimental design, this manuscript does not possess typical results. However, in this context, I would consider strengthening their argument for the development and deployment of this R package. Specifically, I would include a comparison with existing options, including other R packages and functions. The authors must have anticipated such a comparison and include a table in the supplemental material (Table SM1) that provides some of this information, but I would encourage them to summarize key points from this comparison in the text itself. I would also argue that the final statement of the paper (Line 244) isn’t entirely accurate. The datasets and metrics that comprise this paper are mostly if not entirely European. I think the North American audience for this paper could benefit from the inclusion of North American trait data (for example: Vieira N.K.M., Poff N.L., Carlisle D.M., Moulton S.R. II, Koski M. & Kondratieff B.C. (2006) A Database of Lotic Invertebrate Traits for North America, Report Manuscript. United States Geological Survey, Reston, VA. | Poff, N.L., J.D. Olden, N.K.M. Vieira, D.S. Finn, M.P. Simmons, and B.C. Kondratieff. 2006. Functional trait niches of North American lotic insects: traits-based ecological applications in light of phylogenetic relationships. Journal of the North American Benthological Society 25(4):730-755 | Yuan, L. L. Estimation and Application of Macroinvertebrate Tolerance Values (Final). U.S. Environmental Protection Agency, Washington, D.C., EPA/600/P-04/116F/ https://www.epa.gov/risk/freshwater-biological-traits-database-traits ) as well as North American indices and metrics (for example, CEFI; Armanini, D.G., Horrigan, N., Monk, W.A., Peters, D.L. and Baird, D.J., 2011. Development of a benthic macroinvertebrate flow sensitivity index for Canadian rivers. River research and applications, 27(6), pp.723-737). The authors do acknowledge that additional data can be added by the user (Line 138), but inclusion of these data by the authors would reduce the number of users that need to do this independently. I am certainly not expecting this package or this manuscript to be comprehensive, but a reasonable effort could greatly enhance the range of applications for this package. Additionally, I think the authors may miss an opportunity to describe how this set of tools could potentially facilitate greater consistency among biomonitoring programs which could lead to more comprehensive analyses at larger spatial and longer temporal scales.

Additional comments

I agree with the basic premise of this manuscript – that better resources for the efficient and effective analysis of biomonitoring data are urgently needed. It is clear from the presentation of their work that these authors have made significant progress toward this goal with the implementation of their R package for managing ecological data and calculating biomonitoring indices and carefully considered several of the common pitfalls, challenges and obstacles to working with these data. However, there are four essential aspects of the work that I think could be clarified or improved prior to publication, 1) include a brief comparison of biomonitoR to other available tools, especially the scope (this is already included to some degree in the supplemental material but not addressed adequately in the text; for example, Line 223), 2) consider the incorporation of North American datasets and indices (for example, Vieira et al. 2006; Line 138) or indicate an efficient mechanisms for making these types of updates or additions in the future, 3) discuss how tools developed for streamlining data processing could lead to greater consistency and repeatability among studies and how this could enable larger scale or longer-term analyses, and 4) adjust some of the writing to improve clarity. As a final note, I would like to complement the authors on the documents that they made available on github, but recommend they check the spelling of their functions, for example bray_curtys should be bray_curtis. I would also strongly urge the authors to consider compiling a complete R package for submission to the CRAN repository and include this information in the file manuscript if accepted.

Reviewer 3 ·

Basic reporting

Nothing to report – minor changes could be adopted in lines
224 «constrained» not «constraint»
244 «has significant value»
but this is all I spotted.

Experimental design

Nothing to report

Validity of the findings

Nothing to report.

Additional comments

I think this is a well-prepared manuscript, and I have as only suggestion that a language-focused review is conducted by the authors to weed out all inconsistencies (although I believe I found the two last ones).
I congratulate the authors on their manuscript, and especially the R package. I can easily think of several instances in the future where I will need to use functions as provided in biomonitoR.
Also, the R package could be extended to include other indices and metrics as well – and I think the authors will strive to develop biomonitoR even further.

---

## Round 0.2 · accepted · Accept

Thanks for addressing the previous round of reviewer comments; it is clear from the rebuttal letter that these minor issues have been addressed. Congratulations!